# Monoterpene Indole Alkaloids with Ca_v_3.1 T-Type Calcium Channel Inhibitory Activity from *Catharanthus roseus*

**DOI:** 10.3390/molecules26216516

**Published:** 2021-10-28

**Authors:** Zhen-Tao Deng, Wen-Yan Li, Lei Wang, Zhi-Ping Zhou, Xing-De Wu, Zhong-Tao Ding, Qin-Shi Zhao

**Affiliations:** 1Key Laboratory of Medicinal Chemistry for Natural Resource, Ministry of Education and Yunnan Province, School of Chemical Science and Technology, Yunnan University, Kunming 650091, China; dengzhentaoa@mail.kib.ac.cn (Z.-T.D.); zhouzhiping@mail.kib.ac.cn (Z.-P.Z.); 2State Key Laboratory of Phytochemistry and Plant Resources in West China, Kunming Institute of Botany, Chinese Academy of Sciences, Kunming 650201, China; liwenyan@mail.kib.ac.cn (W.-Y.L.); laralei@163.com (L.W.); 3University of Chinese Academy of Sciences, Beijing 100049, China; 4Key Laboratory of Ethnic Medicine Resource Chemistry, State Ethnic Affairs Commission and Ministry of Education, Yunnan Minzu University, Kunming 650500, China

**Keywords:** *Catharanthus roseus*, monoterpene indole alkaloid, catharanosine A, Ca_v_3.1 low voltage-gated calcium channel (LVGCC)

## Abstract

*Catharanthus roseus* is a well-known traditional herbal medicine for the treatment of cancer, hypertension, scald, and sore in China. Phytochemical investigation on the twigs and leaves of this species led to the isolation of two new monoterpene indole alkaloids, catharanosines A (**1**) and B (**2**), and six known analogues (**3**–**8**). Structures of **1** and **2** were established by ^1^H-, ^13^C- and 2D-NMR, and HREIMS data. The absolute configuration of **1** was confirmed by single-crystal X-ray diffraction analysis. Compound **2** represented an unprecedented aspidosperma-type alkaloid with a 2-piperidinyl moiety at C-10. Compounds **6**–**8** exhibited remarkable Ca_v_3.1 low voltage-gated calcium channel (LVGCC) inhibitory activity with IC_50_ values of 11.83 ± 1.02, 14.3 ± 1.20, and 14.54 ± 0.99 μM, respectively.

## 1. Introduction

Monoterpene indole alkaloids (MIAs) are one of the largest natural product families constructed from indole and monoterpene moieties, and commonly found in Apocynaceae, Rubiaceae, and Loganiaceae families [1]. To date, more than 3000 MIAs have been reported, many of which have been found to exhibited important pharmaceutical effects [2,3]. Representative MIAs such as, reserpine, vinblastine/vincristine, and quinine are used clinically for the treatment of hypertension, cancer, and malaria, respectively [4]. In light of their diverse and complex structures and high druggability, MIAs have attracted great interest from chemical and pharmacological communities and have been a potent resource for new drug discovery.

*Catharanthus roseus* (L.) G. Don (Apocynaceae), a tropical perennial subshrub, is a well-known traditional herbal medicine for treating cancer, hypertension, scald, and sore in China [5]. Early phytochemical studies on this plant have led to the isolation of an array of MIAs, including the well-known anticancer drugs vinblastine and vincristine [6]. The discovery of these two drugs has been regarded as one of the most important developments in both natural product chemistry and the clinical treatment of cancer during the 1960s to 1980s [7,8,9,10]. In recent years, some new and bioactive MIAs were still reported from this plant [11,12,13,14,15,16,17]. In our continuing search for structurally unique and pharmaceutically interesting MIAs from medicinal plants [18,19,20,21,22], a phytochemical study of the twigs and leaves of *C. roseus* was undertaken and led to the identification of two new MIAs, catharanosines A (**1**) and B (**2**), and six known analogues (Figure 1). Compound **2** was found to represent an unprecedented aspidosperma-type alkaloid with a piperidine moiety at C-10. Due to the limited amount of **1** and **2**, only compounds **3**–**8** were screened for their inhibitory activity on Ca_v_3.1 low voltage-gated calcium channel (LVGCC), an important therapeutic target for cardiovascular disease [23]. Herein, the isolation, structure determination, and bioactivities of compounds **1**–**8** are described.

## 2. Results

### 2.1. Structure Elucidation

The total crude alkaloid fraction was subjected to MCI gel, silica gel, and Sephadex LH-20 to afford two new MIAs, catharanosines A (**1**) and B (**2**), and six known analogues. Compared with literature data, the known compounds were identified as (*R*)-19-hydroxytabersonine (**3**) [24], lochnerine (**4**) [25], normacusine B (**5**) [26], vincapusine (**6**) [27], vincarodine (**7**) [28], and serpentine (**8**) [29], by comparing their spectroscopic data with those reported in the literature.

Compound **1**, colorless crystals, had a molecular formula of C_20_H_24_N_2_O_3_ according to the ^13^C NMR data and the HR-EI-MS ion at *m/z* 340.1782 [M]^+^ (calcd. 340.1787), which corresponded to 20 indices of hydrogen deficiency. The IR spectrum (Appendix A) displayed absorption bands resulting from hydroxy (3439 cm^−1^), carbonyl (1712 cm^−1^), and aromatic (1630 and 1465 cm^−1^) functionalities. The presence of an oxindole chromophore was revealed by the UV absorptions (Appendix A) at 205, 248, 298 nm [30]. The ^1^H NMR spectrum displayed resonances (Table 1 and Appendix A) for a 1,2,4-trisubstituted benzene ring (*δ*_H_ 6.59 dd, *J* = 8.4, 2.4 Hz, H-11; 6.65 d, *J* = 8.4 Hz, H-12; and 6.76 d, *J* = 2.4 Hz, H-9), an olefinic proton (*δ*_H_ 5.09 q, *J* = 6.7 Hz, H-19), an oxymethylene (*δ*_H_ 3.34, m, H-17), a methyl group (*δ*_H_ 1.40 d, *J* = 6.7 Hz, H_3_-18), and a methoxy group [*δ*_H_ 3.63 (s)]. The ^13^C NMR spectrum (Appendix A), with the aid of the HSQC data (Appendix A), showed 20 carbon resonances (Table 1) attributable to two methyls (one methoxy at *δ*_C_ 55.7), four methylenes (one oxygenated at *δ*_C_ 65.0), eight methines (three aromatic at *δ*_C_ 109.7, 111.8, and 114.9 and one olefinic at *δ*_C_ 114.6), and six nonprotonated carbons (three aromatic at *δ*_C_ 132.0, 135.3, and 154.9, one olefinic at *δ*_C_ 135.6, and one lactam carbonyl at *δ*_C_ 184.4). Comparison of the ^1^H and ^13^C NMR data (Table 1) of **1** with those of rauvomitorine V, an affinisine oxindole type alkaloid from *Rauvolfia vomitoria* [31], revealed their structural similarity, except the absence of one methoxy group and that one aromatic methine in **1** replaced one aromatic nonprotonated carbon in the latter. The only methoxy group was located at C-10 through HMBC correlations (Figure 2) of H-9 (*δ*_H_ 6.76, d, *J* = 2.4 Hz) with C-7 (*δ*_C_ 57.4), C-8 (*δ*_C_ 132.0), C-10 (*δ*_C_ 154.9), and C-13 (*δ*_C_ 135.3), as well as cross-peaks of H-9 and H-11 with the methoxy group in ROESY spectrum (Figure 2).

The relative configuration of **1** was assigned via the ROESY (Appendix A) and ^13^C NMR data (Table 1). The configuration of the spirocyclic C-7 was assigned as *S* by the ROESY correlations (Figure 2) of H-9 with H-6*β*, H-14*β*, and H-16, coupled with the diagnostic chemical shifts of C-2 (*δ*_C_ 184.4), C-3 (*δ*_C_ 62.8), and C-8 (*δ*_C_ 132.0) [32,33]. The ROESY correlations of H-3 and H-5 with H_2_-21, of H-5 with H_2_-17, and of H-9 with H-16 indicated the *α*-orientations of H-3, H-5, and H_2_-17. The rigid structure of the bridge ring system required H-15 to be *α*-oriented. Moreover, the *E*-geometry for the double bond between C-19 and C-20 was determined by the ROESY cross-peak of H-15 and H_3_-18. Finally, single-crystal X-ray diffraction analysis of **1** using Cu-Kα radiation unambiguously assigned the absolute configuration as (3*S*, 5*S*, 7*S*, 15*R*, 16*R*) based on a Flack parameter of 0.11(8) (Figure 3). Accordingly, the structure of **1** was deduced and named as catharanosine A. Many sarpagine type alkaloids have been reported from natural resources [34,35,36,37,38], however, only a few of the corresponding oxindoles were discovered, including affinisine, talpinine, and chitosenine types oxindole alkaloids [31,39,40,41,42]. To date, no general scaffold name for this small group of oxindole alkaloid are reported, so we tentatively name them as spiroindoxyl sarpagane alkaloid.

The HREIMS spectrum of **2** showed a molecular ion peak at *m/z* 539.3000 [M]^+^ (calcd. 539.2995), consistent with a molecular formula of C_30_H_41_N_3_O_6_, suggesting 24 indices of hydrogen deficiency. The IR spectrum (Appendix A) exhibited absorptions attributable to hydroxy (3432 cm^−1^) and carbonyl (1741 cm^−1^) groups, and a benzene ring (1621, 1506, and 1453 cm^−1^). The characteristic UV absorptions (Appendix A) at 213, 261, 309 nm indicated a dihydroindole chromophore [43]. In ^1^H NMR spectrum (Appendix A and Table 1), two aromatic singlets at *δ*_H_ 7.36 (s, H-9) and 5.97 (s, H-12) were observed for the 10,11-disubstituted dihydroindole moiety. The ^13^C NMR spectrum (Appendix A and Table 1) showed the presence of five methyls (one N-methyl, one acetyl methyl, and two methoxy groups), four methylenes, seven methines (one oxygenated at *δ*_C_ 76.2, two aromatic at *δ*_C_ 92.3 and 122.1, and two olefinic at *δ*_C_ 124.4 and 130.0), and nine nonprotonated carbons (one oxygenated at *δ*_C_ 79.4, two ester carbonyl at *δ*_C_ 170.8 and 171.8 and four aromatic at *δ*_C_ 114.4, 124.2, 153.8, and 158.0). In addition, the carbon resonances (Table 1) at *δ*_C_ 54.6, 28.7, 23.4, 22.0, and 45.7, along with the ^1^H-^1^H COSY correlations (Figure 4) of H-2′/H_2_-3′/H_2_-4′/H_2_-5′/H_2_-6′, indicated the presence of 2-piperidinyl unit in **2**. The ^1^H and ^13^C NMR spectroscopic data (Table 1) of **2** closely resembled those of vindoline [44], with the exception of the aforementioned 2-piperidinyl moiety. The HMBC correlations (Figure 4) from H-9 (*δ*_H_ 7.36) to C-7 (*δ*_C_ 52.8), C-8 (*δ*_C_ 124.2), C-10 (*δ*_C_ 114.4), and C-13 (*δ*_C_ 153.8), from H-2′ (*δ*_H_ 4.29, br s) to C-9 (*δ*_C_ 122.1), C-10, and C-11 (*δ*_C_ 158.0), as well as the cross-peak of H-12 with N-methyl and 11-methoxy groups, revealed the linkage of vindoline with piperidine moiety through C-10-C-2′ bond.

The ROESY correlations (Figure 4) of H-2/H-6*β* and H-6*β*/H-3*β* suggested the *β*-orientation of H-2 and the *R** configuration for C-7. H-17, H-21, and the ethyl groups were determined as α-orientated by the ROESY cross-peaks of H-21/H_3_-18 and H-17/H_2_-19. The *β*-orientation of 16-OH was established by similar carbon resonances at C-2 (*δ*_C_ 83.3; Δ*δ*_C_ +0.1), C-16 (*δ*_C_ 79.4; Δ*δ*_C_ −0.1), and C-17 (*δ*_C_ 76.2; Δ*δ*_C_ +0.0) with those of vindoline [44]. However, other ROESY correlations for **2** were insufficient to determine the configuration at C-2′. Consequently, the structure of compound **2** was identified and named as catharanosine B. To our knowledge, compound **2** represented the first aspidosperma-type alkaloid with a 2-piperidinyl moiety at C-10. Biogenetically, compound **2** was likely to be formed through electrophilic substitution at C-10 of vindoline by Δ^1^-piperidinium cation derived from *L*-lysine via decarboxylation and oxidation under the catalysis of lysine decarboxylase and amine oxidase, and subsequently cyclization and protonation (Figure 1) [45,46].

### 2.2. Biological Activity

Due to the traditional use of *C. roseus* for treating hypertension in China, compounds **3**–**8** were evaluated for the effects on Ca_v_3.1 low-voltage-gated calcium channel, which plays an important role in the regulation of cardiovascular disease. At a concentration of 50 μM, compounds **6**–**8** showed strong inhibitions on Ca_v_3.1 (Figure 5), while compounds **3**–**5** exhibited weak activity with inhibition rate of less than 50%. Then, compounds **6**–**8** were further evaluated for their dose-dependent relationships on Ca_v_3.1 at a concentration range from 1.6 to 50.0 μM. The results showed that compounds **6**–**8** dose-dependently inhibited on Ca_v_3.1 with IC_50_ values of 11.83 ± 1.02, 14.30 ± 1.20, and 14.54 ± 0.99 μM, as compared to mibefradil, an inhibitor of T-type VGCC, with IC_50_ value of 3.09 ± 0.41 μM (Figure 6). These results indicated that compounds **6**–**8** were important antihypertensive active components of *C. roseus*.

## 3. Materials and Methods

### 3.1. General

Optical rotations were measured with a Horiba SEPA-300 polarimeter (Horiba, Tokyo, Japan). Melting point was recorded on an X-4 micro melting point apparatus (Beijing Second Optical Instrument Factory, Beijing, China). IR spectra were obtained by a Tensor 27 spectrophotometer (Bruker, Karlsruhe, Germany) with KBr pellets. UV spectra were obtained using a Shimadzu UV-2401A spectrophotometer (Shimadzu, Kyoto, Japan). 1D and 2D spectra were run on a Bruker AM-400 or an Avance III 600 spectrometer (Bruker, Karlsruhe, Germany) with TMS as the internal standard. Chemical shifts (*δ*) were expressed in ppm with reference to the solvent signals. EIMS were recorded on a Waters Autospec Premier P776 spectrometer (Waters Corporation, Milford, MA, USA). Column chromatography (CC) was performed using silica gel (200–300 mesh, Qingdao Marine Chemical Co., Ltd., Qingdao, China) and MCI gel (75–150 mm; Mitsubishi Chemical Corporation, Tokyo, Japan). Fractions were monitored by TLC (GF254, Qingdao Marine Chemical Co., Ltd., Qingdao, China), and spots were visualized by heating silica gel plates sprayed with 10% H_2_SO_4_ in EtOH. All solvents were distilled prior to use.

### 3.2. Plant Material

The whole plants of *C. roseus* were purchased from the Herb Material Market of Juhuacun, Kunming, Yunnan Province, P. R. China, in June 2011, and identified by Prof. Xiao Cheng, Kunming Institute of Botany, Chinese Academy of Sciences. A voucher specimen (20110620C) was deposited at the State Key Laboratory of Phytochemistry and Plant Resources in West China, Kunming Institute of Botany, Chinese Academy of Sciences.

### 3.3. Extraction and Isolation

The air-dried whole plants of *C. roseus* (60 kg) were powdered and extracted with methanol (4 × 200 L) under reflux. The methanol was evaporated under reduced pressure to produce a residue, which was dissolved in hot water and adjusted to pH 2 with 0.5% HCl and then extracted with ethyl acetate (50 L × 3). The water-soluble portion was adjusted to pH 9.0 with sat. Na_2_CO_3_ and partitioned with CHCl_3_ to yield the total crude alkaloids (200 g), which were chromatographed over silica gel column using a step-gradient eluting with CHCl_3_-Me_2_CO (1:0-0:1) to obtain five fractions A–E. Fraction B was applied to MCI gel column eluted with MeOH-H_2_O (40%-100%) to give subfractions B1–B4. Fraction B1 was subjected to silica gel CC (CHCl_3_-MeOH, 10:1) to obtain compounds **6** (15 mg), **7** (15 mg), and **8** (12 mg). Fraction B2 was subjected to silica gel CC (CHCl_3_-MeOH, 10:1) to obtain compounds **3** (13 mg) and **4** (11 mg). Fraction C was subjected to silica gel CC (CHCl_3_-MeOH, 9:1) to obtain compound **5** (5 mg). Fraction D was applied to MCI gel column eluted with MeOH-H_2_O (30:70, 40:60, 50:50, 60:40, 70:30, 80:20, 90:10, 100:0, each 3 L) to give subfractions D1-D3. Fraction D1 was subjected to silica gel CC (CHCl_3_-MeOH, 20:1) and then purified by Sephadex LH-20 (CHCl_3_-MeOH, 1:1) to obtain compound **1** (2 mg). Compound **2** (1.5 mg) was obtained by Sephadex LH-20 (MeOH) from Subfraction D3.

### 3.4. Spectroscopic Data of Compounds **1** and **2**

Catharanosine A (**1**): Colorless crystals, mp 104−108 °C; ${\left[\alpha \right]}_{D}^{23}$-21.80 (*c* 0.12, MeOH); UV (MeOH): *λ*_max_ (log*ε*) 298 (2.61), 248 (3.04), 205 (3.64) nm; IR (KBr) *υ*_max_: 3439, 2932, 1712, 1630, 1465, 1206 cm^−1^. ^1^H and ^13^C NMR spectral data, see Table 1; HR-EI-MS *m/z* 340.1782 (calcd. for C_20_H_24_N_2_O_3_, 340.1787).

Catharanosine B (**2**): Yellow oil; ${\left[\alpha \right]}_{D}^{23}$-98.25 (*c* 0.14, MeOH); UV (MeOH): *λ*_max_ (log*ε*) 309 (3.29), 261 (3.52), 213 (3.99) nm; IR (KBr) *υ*_max_: 3432, 2937, 1741, 1621, 1506, 1453, 1433, 1383, 1231, 1041 cm^−1^. ^1^H and ^13^C NMR spectral data, see Table 1; HR-EI-MS *m/z* 539.3000[M + H]^+^ (calcd. for C_30_H_41_N_3_O_6_, 539.2995).

### 3.5. X-ray Crystal Data of **1**

C_20_H_24_N_2_O_3_·H_2_O, *M* = 358.43, *a* = 12.2146(2) Å, *b* = 12.2146(2) Å, *c* = 24.9041(6) Å, *α* = 90°, *β* = 90°, *γ* = 90°, *V* = 3715.60(15) Å^3^, *T* = 100(2) K, space group *P*41212, *Z* = 8, *μ*(CuKα) = 0.727 mm^−1^, 17875 reflections measured, 3394 independent reflections (*R_int_* = 0.1146). The final *R*_1_ values were 0.0651 (*I >* 2*σ(I)*). The final *wR*(*F*^2^) values were 0.1599 (*I* > 2*σ*(*I*)). The final *R_1_* values were 0.0654 (all data). The final *wR*(*F*^2^) values were 0.1603 (all data). The goodness of fit on *F*^2^ was 1.125. Flack parameter = 0.11(8). Crystallographic data for compound **1** have been deposited in the Cambridge Crystallographic Data Centre (deposition numbers: CCDC 2106217). Copies of these data can be obtained free of charge via www.ccdc.cam.ac.uk.

### 3.6. Ca_v_3.1 T-Type Calcium Channel Inhibitory Activity Assay

HEK293T cells purchased from ATCC were cultured at 37 °C with 5% CO_2_ in Dulbecco’s modified Eagle medium with glucose, L-glutamine, pyruvate, 10% FBS, and 1% Pen-Strep. Cells were seeded at low density onto 24-well plates 24 h before transfection. Adherent cells were transfected using Lipofectamine 2000 reagent (Invitrogen) with 300 ng Ca_v_3.1 cDNA and recorded after 48 h. Whole-cell voltage-clamp recordings were performed at room temperature (24 °C). The peak currents of Ca_v_3.1 were elicited by 150 ms depolarization from a holding potential of −100 mV to −40 mV at 4 s intervals. Borosilicate glass micropipettes were pulled to produce a resistance of 4–6 MΩ and filled with intracellular recording solution containing 130 mM CsCl, 2 mM MgCl_2_, 10 mM EGTA, 5 mM Na-ATP, 10 mM HEPES (pH 7.2 with CsOH). The extracellular recording solution was composed of 145 mM CsCl, 1 mM MgCl_2_, 2 mM CaCl_2_, 10 mM glucose, 10 mM HEPES (pH 7.4 with CsOH). The current trace of Ca_v_3.1 in different states was analyzed by the Clampfit 10.6. Data were processed using the software Graphpad Prism 8.0.

## 4. Conclusions

In summary, two new MIAs, catharanosine A (**1**) and catharanosine B (**2**), together with six known compounds (**3**–**8**) were isolated from the twigs and leaves of *Catharanthus roseus*. The absolute configuration of compound **1** was confirmed by X-ray crystal diffraction analysis. Compound **2** represented the first aspidosperma-type alkaloid with a 2-piperidinyl moiety at C-10. Compounds **6**–**8** dose-dependently inhibited on Ca_v_3.1 with IC_50_ values of 11.83 ± 1.02, 14.30 ± 1.20, and 14.54 ± 0.99 μM.

## Data Availability

The data presented in this study are available in Appendix A.

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
