# Peer review of "Monoterpene Indole Alkaloids with Cav3.1 T-Type Calcium Channel Inhibitory Activity from Catharanthus roseus"

_molecules, 2021, doi:10.3390/molecules26216516_

Round 1

Reviewer 1 Report

The present manuscript deals with the isolation and structure elucidation of 2 new monoterpene indole alkaloids from the thoroughly-studied species Catharanthus roseus, Apocynaceae. Six additional MIA had also been isolated and only these latter could be assayed for their bioactivities, viz. low voltage-gated calcium channels.

The compounds were nicely isolated and the spectroscopic data are good.

I'm unsure whether deserves or not to be published as I have many significant concerns with this manuscript. In any case, it requires major improvements it this draft is further considered for publication. These concerns are primarily related to both the scientific content of the article but I would also like to encourage the authors to present their results in a less raw manner. Are the newly reported compounds affiliated to formerly reported compounds from the same plant ? Which biosynthetic scenarios can be envisaged to account for the piperidinyl group in C-10 which is claimed to be new etc. ? Likewise, the outcomes of the biological assays are not even being discussed. Here are some thoughts to remedy just below:

  • as a phytochemist involved in the description of new MIAs myself, I was bewildered to see how the structure elucidation part of the manuscript was written. In particular, the scaffold of the first newly reported structure is not even being reported ! This reviewer would have appreciated it to be named: it is a spiroindoxyl sarpagane. The elucidation of this structure does not raise much concern given that the authors benefit from an ORTEP cliché, yet, some discussed ROESY features interpretations are quite unclear to me. What does a S* designation means when a single stereocenter is being discussed ? It litterally means that it could either be S or R when applied to a single stereocenter => please revise ! Besides, this puzzling interpretation of ROE correlations guide the authors to determine a beta-orientation to H-15 which is inconsistent with the drawing of the structure and which would be very surprising given that sarpaganes involve a non-rearranged secologanin unit, fixing C-15 configuration as S (like in strictosidine), meaning alpha oriented as drawn here. I would suggest the authors to avail themselves of the configuration of chitosenine which indeed depicts a correct sarpagane configuration. If I were the authors, I would initiated the configurational aspects of the manuscript indicating that as a sarpagane, C-15 configuration is fixed, to further spread relative configuration information through ROE crosspeaks...
  • I have considerable difficulties with the structure elucidation of 2, and if the authors are not able to go any further in the process, I would suggest to remove this molecule from the manuscript. Unlike compound 1, compound 3, as an Aspidosperma-type alkaloid may exist in any enantiomeric series so that the lack of any spectroscopic data to assign an absolute configuration represents a genuine issue. Part of the relative configuration related to the monoterpenic component  has been addressed. Yet, direct evidence does not appear for the assignment of C-16 configuration. The authors state that it is quite similar to vindoline, which may indeed be a sound argument, yet, references lack to convince me that the other epimeric series would have resulted in different NMR landmarks. A great way to address this issue would have been to perform DFT-NMR and subsequent DP4 probability scores to assign an epimeric series with quantifiable confidence ! The same applies to the relative configuration of C-1' (mistakenly referred to as C-2' in the manuscript) which cannot benefit from ROESY NMR to be addressed. Besides, although the paper is very much based on the unprecedented occurrence of a piperidine in its structure, no hypothesis is being provided to account for it ! Could you provide further clues indicating that this molecule occurs as a genuine natural product ? Could it be related somehow to a step in the extraction/isolation process ? Likewise, it is stated that this the first piperidine-containing aspidosperma but I would have been interested to know whether other structural types of MIA already comprised this additional ring or not on their indolic component ?
  • Biological evaluation: it is quite surprising to me not to have a positive control indicating that the assay proceeds conveniently. Likewise, as I am not an expert of this text, I have no idea whether these results are interesting or not. Again, I assume that the results are presented in a very raw manner.
  • At last, I would suggest the authors to carefully proofread themselves, there are many nasty mistakes throughout the document. Here are a few examples which I spotted when going through the manuscript.

"2-piperidiny unit" appears twice => do you mean 2-piperidinyl instead.

"However, other ROESY correlations for 2 were insufficient to determined the configuration at C-2'." => should read determine

Overall, this manuscript deserves at least major revisions to overcome the numerous inconsistencies spotted when reporting on compound 1. Regarding 2, I think that in 2021, it is no longer acceptable to publish new MIAs without a fully determined structure (that is an absolute configuration), as the compound does not even comprise a relative configuration, I would suggest to suppress this compound from the manuscript but I am unsure that the overall merit is enough to deserve publication in Molecules.

Reviewer 2 Report

The manuscript describe the isolation and structure elucidation of two new alkaloids (1-2) and six known derivatives (3-8) from a Chinese plant, Catharanthus roseus. The new structures were confirmed by 2D NMR and a single crystal x-ray data for 1. In addition, the known compounds were tested for their potential in inhibiton against voltage gated calcium channel. 

1) The configuration of piperidine at C-1' in 2 was not determined.

2) Is there a chance that compound 2 was an artifact by exposing to piperidine? Are there any known example for this moiety in natural products?

Author Response

1. The configuration of piperidine at C-1' in 2 was not determined.√ We tried to perform DFT-NMR and subsequent DP4 probability scores to determine the configuration of piperidine at C-2' according to the suggestion of reviewer 1. However, we found that the consistence of the calculated data was not so good and large computational errors can be observed. With our confidence in the proposed structure, we think that the current computational method might have limitations in reproducing the experimental NMR chemical shifts of 2. We think that DFT-NMR and subsequent DP4 probability scores are important methods to determine the structures and configurations of natural products, but it could not solve the problem of all compounds. We will try to solve the configuration of C-2' by X-ray diffraction analysis if we isolate compound 2 from Catharanthus roseus in the future study.2. Is there a chance that compound 2 was an artifact by exposing to piperidine? Are there any known example for this moiety in natural products?√ We found two analogues of compound 2, vindogentianine (aspidosperma-type alkaloid with gentianine) [Fitoterapia 2015, 102, 182–188.] and bannucine (aspidosperma-type alkaloid with pyrrolidone) [Tetrahedron, 2015, 71, 9579-9586] were found from the same plant (Catharanthus roseus). From this point of view, compound 2 should be natural product. In addition, lycoplatyrine A, one lycodine-type alkaloid with a piperidine moiety was found from Lycopodium platyrhizoma [J. Nat. Prod. 2019, 82, 324−329.].
